# A Survey of Deep Learning for Electronic Health Records

Jiabao Xu [1,†], Xuefeng Xi [1,*,†], Jie Chen [2], Victor S. Sheng [3,*], Jieming Ma [4] and Zhiming Cui [1]

1   School of Electronic and Information Engineering, Suzhou University of Science and Technology, Suzhou 215009, China
2   School of Cyber Science and Engineering, Southeast University, Nanjing 211100, China
3   Department of computer science, Texas Tech University, Lubbock, TX 79409, USA
4   School of Advanced Technology, Xi'an Jiaotong-Liverpool University, Suzhou 215000, China
*   Correspondence: xfxi@usts.edu.cn (X.X.); victor.sheng@ttu.edu (V.S.S.)
†   These authors contributed equally to this work.

**Abstract:** Medical data is an important part of modern medicine. However, with the rapid increase in the amount of data, it has become hard to use this data effectively. The development of machine learning, such as feature engineering, enables researchers to capture and extract valuable information from medical data. Many deep learning methods are conducted to handle various subtasks of EHR from the view of information extraction and representation learning. This survey designs a taxonomy to summarize and introduce the existing deep learning-based methods on EHR, which could be divided into four types (Information Extraction, Representation Learning, Medical Prediction and Privacy Protection). Furthermore, we summarize the most recognized EHR datasets, MIMIC, eICU, PCORnet, Open NHS, NCBI-disease and i2b2/n2c2 NLP Research Data Sets, and introduce the labeling scheme of these datasets. Furthermore, we provide an overview of deep learning models in various EHR applications. Finally, we conclude the challenges that EHR tasks face and identify avenues of future deep EHR research.

**Keywords:** electronic health records (EHR); machine learning (ML); deep learning; de-identification; privacy preservation; deep EHR; natural language processing (NLP)

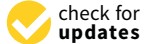

## 1. Introduction

In recent years, the use of Electronic Health Record (EHR) systems by healthcare organizations worldwide have increased dramatically. Compared to paper-based medical record management, electronic medical records are easier to store and use. Along with the yearly increase in the adoption of EHR systems, medical data is also growing significantly. This data often contains a wealth of medical information. Although EHRs were originally designed to archive medical information, researchers have found that this type of information can provide effective support in a variety of clinical informatics applications.

Early on, traditional machine learning techniques were often used to leverage EHR data. However, it is still not enough to provide an effective and reliable analysis structure. Rapidly developing deep learning provides good technical support for better processing and analysis of such big medical data [1–4]. Especially in the real environment of huge data volume, deep learning techniques based on deep data feature analysis have the ability to provide efficient and stable analysis results [5]. After a period of development, the research of deep learning techniques in EHR has been divided into four areas: Information Extraction, Representation Learning, Medical Prediction, and Privacy Protection. The number of deep learning papers based on EHR is increasing, and it has been able to outperform traditional algorithms and move closer to the expert level.

This survey selected relevant research work retrieved by Google Scholar from 2016, selecting work with good performance or innovative work with inspiring ideas within the same period. It is important to note that in presenting deep learning models and early work, we selected the papers with the highest recognition.

The purpose of this survey is to provide technical references for readers and researchers to advance the research and development of deep learning on EHR. The evolution of EHR is discussed first, followed by a summary of the most recognized EHR datasets and deep learning models in EHR. Following that, it goes through current achievements and algorithm innovation in this area. Finally, this survey will discuss the existing limitations and provide some suggestions for the development of deep learning applications based on EHR.

## 2. Evolution of EHR

In recent years, the use of Electronic Health Record (EHR) systems [6] by healthcare organizations worldwide has increased dramatically. Compared to paper-based medical record management, electronic medical records are easier to store and use. Along with the annual increase in the adoption of EHR systems, medical data is also growing significantly. This data often contains a wealth of medical information. Although EHRs were originally designed to archive medical information, researchers have found that this type of information can provide effective support in a variety of clinical informatics applications.

Initially, the EHR systems were designed to handle the core administrative functionalities of hospitals, allowing for the use of controlled vocabulary and labeling schemes. There are various labeling schemes such as ICD (International Statistical Classification of Diseases) for diagnostic [7,8], CPTta (Current Procedural Terminology) for procedures [9], and LOINC (Logical Observation Identifiers Names and Codes) for laboratories [10], ATC (Anatomical Therapeutic Chemical) for drug [11], and RxNorm for medication [12]. The different labeling schemes result in standard datasets for various specialties. The number of EHR data is increasing year by year as the EHR system improves, and there have been numerous types of research on the secondary use of these data.

Early on, traditional machine learning techniques, such as Support Vector Machine (SVM) [13], Random Forest [14], and Logical Regression, were often used to leverage the rich EHR data. However, it is still not enough to provide an effective and reliable analysis structure. Rapidly developing deep learning provides good technical support for better processing and analysis of such big medical data. Especially in the real environment of huge data volume, deep learning techniques based on deep data feature analysis have the ability to provide efficient and stable analysis results. The number of deep learning papers in this area is increasing, and it has been able to outperform traditional algorithms and move closer to the expert level. This survey checked the number of research papers connected to EHR on *Web of Science* as of 2021, and the number of papers published increased dramatically, as shown in Figure 1. The use of EHR standards when recording medical events and related information. in computer science account for 29.488 percent of the entire application, as illustrated in Table 1. Deep learning, as opposed to typical machine learning approaches, produces better results with less preprocessing time and better prediction results.

**Table 1.** Statistics of research papers on artificial intelligence in the field of medical information.

| Research | Record Numbers | Percentage (%) |
|---|---|---|
| Health Care Sciences Services | 851 | 73.489 |
| Computer Science | 771 | 66.580 |
| Mathematical Computational Biology | 622 | 53.713 |
| Medical Informatics | 455 | 39.292 |
| Mathematics | 379 | 32.729 |

Statistical source: www.webofscience.com (accessed on 28 July 2022).

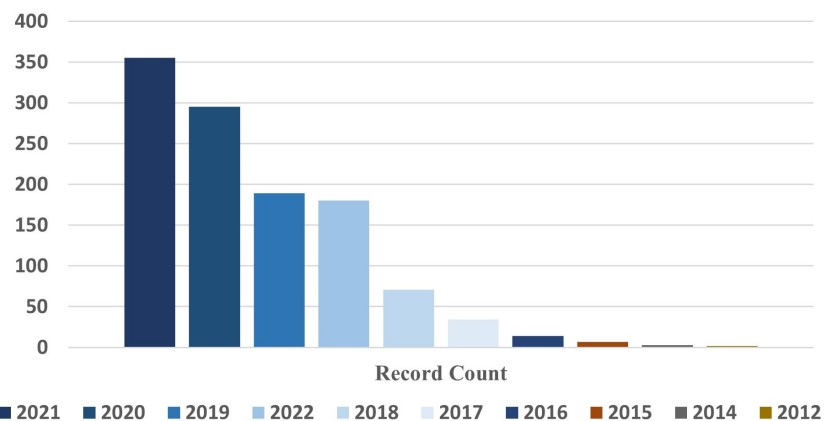

**Figure 1.** Number of papers published on AI algorithms in the EHR field over the past decade. This area is receiving more attention from computer researchers every year.

## 3. EHR Datasets

The EHR covers a large amount of medical and health information and with the rapid development of eHealth, its data volume is growing rapidly. Making full use of EHR data can facilitate medical research and medical knowledge discovery and accelerate the response time of medical services. Currently, several organizations and institutions have conducted development work on EHR datasets, and several open datasets have been formed, such as MIMIC, PCORnet, Open NHS and eICU, etc. The publication of each EHR dataset needs to follow the HIPAA (Health Insurance Portability and Accountability Act) [15]. This section will give a brief introduction to these datasets.

### 3.1. Medical Information Mart for Intensive Care (MIMIC)

The MIMIC-III [16] data set was issued by the Massachusetts Institute of Technology (MIT) in 2016. It has almost 1400 citations and 60,000 ICU hospitalization records. It is the best real-world dataset of in-hospital treatment and monitoring currently available free to researchers worldwide. The structure depicted in Figure 2 is the basic structure of the database.

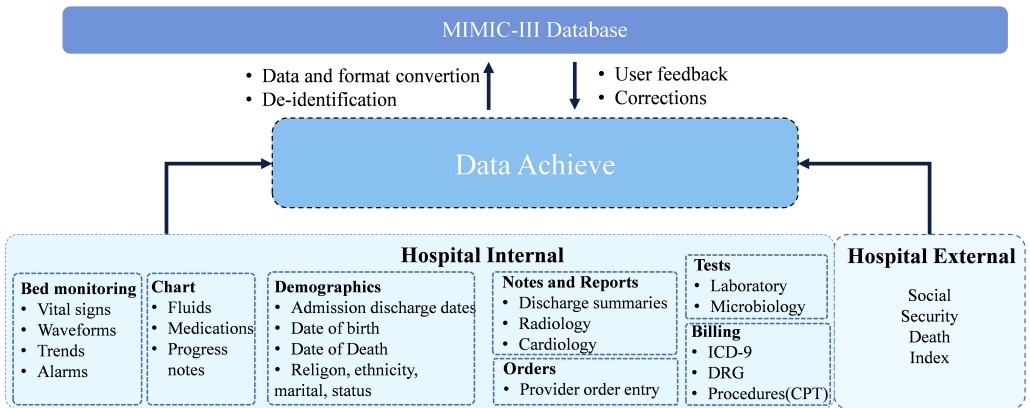

**Figure 2.** MIMIC-III dataset collection and dataset construction schematic.

The MIMIC-III dataset has a large amount of medical information which mainly includes the waveform dataset and the clinical dataset, which can be divided into 21 data tables and 5 dictionary information auxiliary tables according to the record content. The data tables include admissions, callout, caregivers, chartevents, cptevents, datetimeevents, diagnoses_icd, drgcodes, icustays, inputevents_cv, inputevents_mv, labevents, microbiologyevents, noteevents, outputevents, patients, prescriptions, procedureevents_mv,

procedures_icd. Dictionary information auxiliary tables include d_cpt, d_icd_diagnoses, d_icd_procedures, d_items, and d_labitems.

Therefore, users can use both traditional statistical methods to study the relationship between treatments and prognoses, as well as data mining and machine learning algorithms to research corresponding topics. Furthermore, after years of development, MIT has released the MIMIC-IV [17] data set recently, which modularizes the MIMIC-III data table and allows the division structure to represent data independence and differences between modules.

### 3.2. eICU-CRD

The eICU-CRD [18] builds on the success of the MIMIC-III release, a large public dataset created by Philips in collaboration with the MIT Laboratory for Computational Physiology (LCP). Building on MIMIC-III, the dataset extracts data from multiple institutions for research scoping. The dataset collects a large amount of clinical information including vital signs, care plan documents, diagnostic information, etc., and is freely available. It has good usability for many applications such as machine learning algorithms, decision support tool development, and clinical research development.

### 3.3. PCORnet

PCORnet is an EHR dataset created by The Patient-Centered Outcomes Research Institute (PCORI) in the United States in 2013 [19]. The dataset contains medical details on up to 30 million patients and includes 11 clinical data research networks and 18 patient enhancement networks. Researchers have easy and fast access to a large amount of medical data using the official PCORnet Common Data Model (CDM). The CDM has been updated to version v6.0.

### 3.4. Open NHS

The Open National Health Service (NHS) is an open-source dataset created by the NHS in the UK in 2011. Users include healthcare providers, patients, and computer-related organizations and researchers. The dataset currently contains images, text, and other datasets from 13 different medical fields in the health and care sector. The NHS is actively expanding the dataset to help researchers access the latest medical information quickly and up-to-date.

### 3.5. NCBI-Disease

NCBI-disease is a fully annotated corpus of medical texts provided by the National Center for Biotechnology Information (NCBI). NCBI-disease includes both the names of diseases and their matching MeSH or OMIM identities. The dataset consists of four main classes, Modifiers, Specific Mentions, Composite Mentions, and Disease Class, and is primarily used for the disease-named entity recognition studies.

### 3.6. i2b2/n2c2 NLP Research Data Sets

Informatics for Integrating Biology and the Bedside (i2b2) is a national biomedical computing project sponsored by the National Institutes of Health (NIH) [20]. As shown in Figure 3, i2b2 has been publishing various medical shared tasks with related datasets since 2006. As shared tasks, i2b2 provided data with wide reach in the medical NLP community and was one of the first to desensitize EHR data and then make it available to the public. Note that after the i2b2 project ended the name was changed to National NLP Clinical Challenges (n2c2) and continues to be maintained by the Department of Biomedical Information at Harvard Medical School.

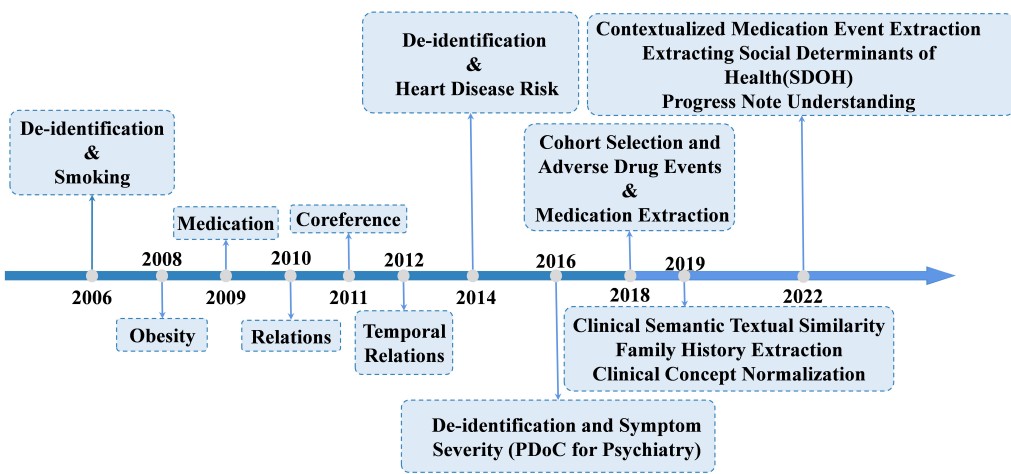

**Figure 3.** Shared tasks published by i2b2/n2c2. Since 2018 the i2b2 project ended the name was changed to n2c2. For each share task, a high-quality dataset has been officially published.

## 4. Deep Learning Models

Deep learning has a wide range of algorithms. This section will provide an overview of deep learning models that are often used in EHR. The survey of deep learning [21] is a full definition and explanation of deep learning for those who want to learn more about it. This survey attempts to describe the key equation and model of each deep learning method, as well as introduce the associated deep learning algorithm.

### 4.1. Artificial Neural Networks (ANNs)

Before going over several deep learning approaches, this survey will go over the architecture of Artificial Neural Networks (ANNs), which is the foundation for most deep learning algorithms. The hierarchical sorting structure depicted in Figure 4 is a simple three-layer ANN made up of the input layer, hidden layers, and output layer, in that sequence. The smallest unit-neuron of the neural network is represented by each circle, and the neurons in different layers are joined to form a neural network. The hidden layer's neuron is also known as a hidden unit. ANN provides a prototype for the construction of other deep models.

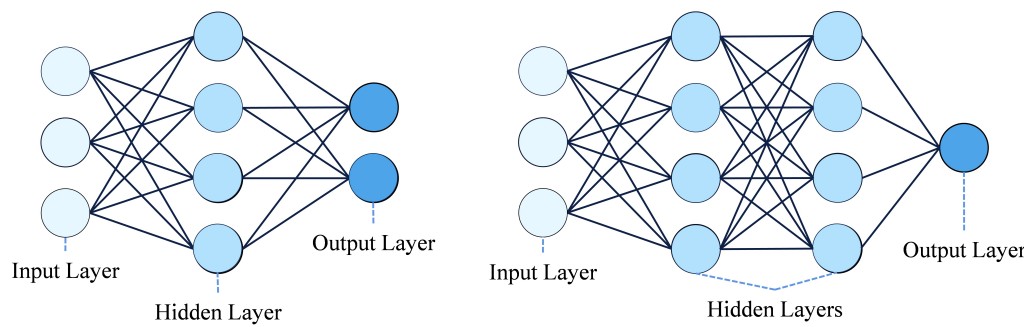

**Figure 4.** Neural network model with single hidden layer on left and double hidden layer on right.

### 4.2. Convolutional Neural Networks (CNN)

The convolutional neural network is a representative deep learning method that was first utilized for image feature extraction. When it utilizes its capacity to extract some features from data, these features potentially contain features that humans cannot perceive. The structure, such as LeNET-5 [22], is often a combination of convolution layers and pooling layers. In addition, there are many derivative algorithms based on CNN architecture, such as AlexNet [23], VGGNet [24], ResNet [25], etc.

In Equation (1), *x* represents the input, and *w* is the weighted function of the equivalent convolution filter for one-dimensional data. From one-dimensional to two-dimensional

data, *X* represents two-dimensional data in Equation (2), and *K* represents the convolution kernel. To obtain the feature map, the convolution process can be seen as the sliding product of the input matrix *X* and the convolution kernel matrix *K*.

$$C_{1d} = \sum_{\alpha=-\infty}^{\infty} x(\alpha)w(t-\alpha) \tag{1}$$

$$C_{2d} = \sum_{m}\sum_{n} X(m,n)K(i-m,j-n) \tag{2}$$

CNN could even extract text features, which is referred to as Text-CNN [26] to distinguish it from CNN in images. The Text-CNN network is the model architecture displayed in Figure 5. Text-CNN takes a pre-trained word vector as input and then generates the appropriate word embedding. In contrast to general CNN, the width of the convolution kernel in convolution must be the same as the word vector's dimension. For example, the convolution kernel in Figure 5 has a dimension of six and is a two-dimensional matrix. The pooling and other procedures that follow are similar to those used by CNN in general.

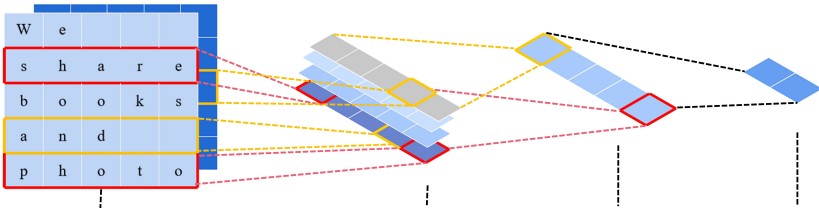

N×K representation of sentence with two channels (static and non-static)    Convolutional layer with multiple filter widths and feature maps    Max-over time pooling    Fully connected layer with dropout and softmax output

**Figure 5.** Two-channel Text-CNN network intent. After the convolution layer, it also traverses the linear layer. The input to the network is a representation matrix $X \in \mathbb{R}^{N \times K}$. Where N is the number of characters in a word and K is the number of characters in a word that contains the most characters.

When it comes to extracting text features in EHR, Text-CNN technology plays a critical role. Many technologies are aimed at optimizing the medical text features obtained by Text-CNN, reducing the privacy information features of patients, highlighting the features useful for downstream tasks, and achieving the effect of protecting private information, in order to obtain the effect of protecting private information.

### 4.3. Recurrent Neural Networks (RNN)

CNN is frequently used to process time series or text language, and it is usually utilized to process input data or time series with a clear spatial structure. Although CNN can process time series, it is substantially weaker than RNN when compared to the feature information retrieved by RNN, which means it is better at processing data with long-term time dependence. To put it another way, RNN is more concerned with the overall feature representation of data.

The actual frame of the RNN is presented on the left, and its expanded version is given on the right, as seen in Figure 6. The network can be seen to be a sequence structure, which has the advantage of preprocessing data of sequence types but also exposes the network to the risk of gradient explosion and disappearance.

The RNN network has numerous difficulties in the practical application of EHR, such as gradient explosion or gradient disappearance. As a result, Long Short-Term Memory (LSTM) [27] and Gated Recurrent Neural Networks (GRNNs) [28] are commonly employed in practice. RNN is used in the development of both of them. The common thread is that while updating on $[t-1, t]$, the addition operation is used to reduce the occurrence of gradient difficulties.

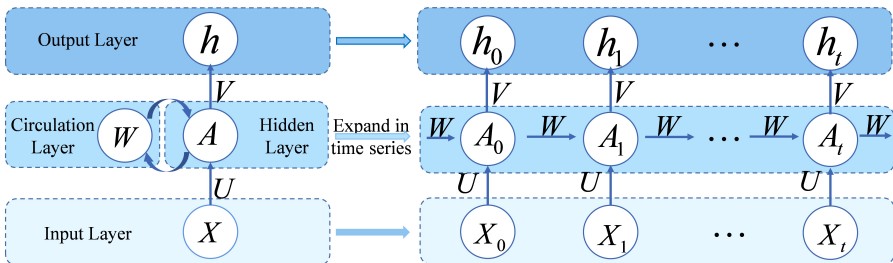

**Figure 6.** The structure of RNN.

The gradient problem is solved by translating the gradient descent product into addition in both LSTM and Gated Recurrent Neural Networks. When comparing the network models of the Gated Recurrent Neural network with LSTM, the Gated Recurrent Neural network has fewer parameters and is easier to converge, but LSTM still outperforms the Gated Recurrent Neural network when dealing with huge data sets. Users must make a reasonable decision based on their own requirements.

*4.4. Auto-Encoder (AE)*

Auto-Encoder (AE) is an unsupervised learning method, unlike neural networks, which utilize a huge number of data sets for end-to-end training to increase the network's accuracy, in which the data entering the input layer and exiting the output layer is extremely similar to the data entering the input layer. The input data is normally encoded by the encoding network in this procedure, and subsequently identical outcomes are produced by decoding. The encoding function is $f(x)$, the decoding function is $g(x)$, and the constraint condition is $x \approx x'$, as illustrated in the network topology in Figure 7.

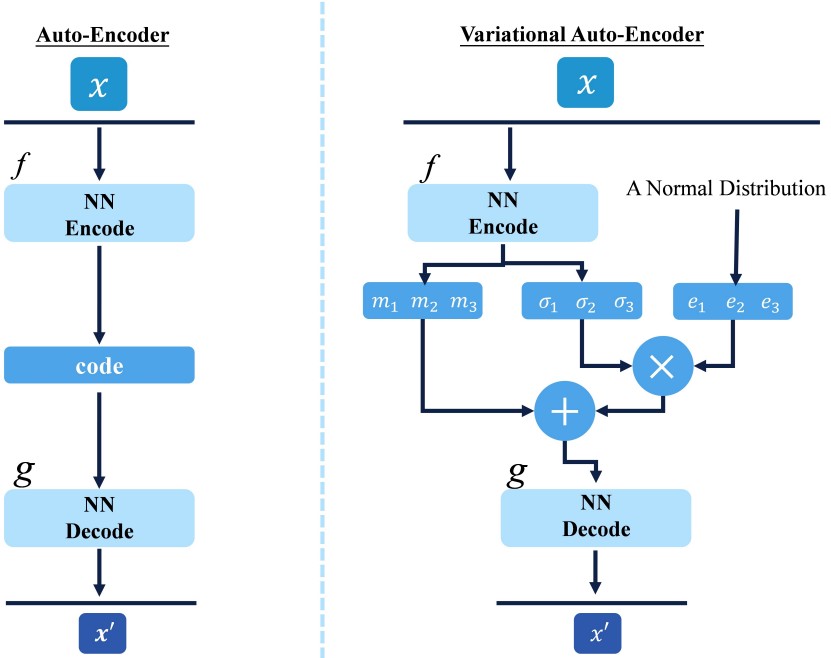

**Figure 7.** The comparison chart between AE and VAE.

It is frequently utilized as a variant network of AE, namely Variational Auto-Encoder, in medical health records. The Variational Auto-Encoder (VAE), in comparison to the Auto-Encoder, tends to create data. After the network has been trained, the data generated by the conventional normal distribution can be fed into the decoder to generate new data that is similar to but distinct from the training set, similar to the generating confrontation network described below. VAE is more complicated in the network model than AE and

refines the section between encoder and decoder in AE such that the network gets a given distribution of the original data, as seen in Figure 7. It is through this optimization that VAE is able to generate data. Currently, several researchers are using the network's properties to improve the original RNN network, such as the Variational Recurrent Neural Network (VRNN) network [29], which modifies the vanilla AE to allow the network to dig out possible information in the data.

### 4.5. Generative Adversarial Network (GAN)

Goodfellow proposed the GAN [30] network model. In recent years, GAN has excelled in a variety of areas, including text-image generation, image superresolution, and so on. GAN, like VAE, may also produce data. As shown in Figure 8, the generative and discriminant models make up the majority of the GAN model. Generator (G) collects the distribution of sample data and generates noise that fits a specific distribution to produce a sample that looks like the actual thing. The generator's output is received by the discriminator (D), which is a two-classifier. The output probability is great when the fake data generated by G is comparable to the genuine data; otherwise, the output probability is minimal. With that example, the discriminator will score higher if the generated data is more comparable to the training data. Equations (3) and (4) depict the alternating training process. There are also many variants of GAN networks, such as CGAN [31], DCGAN [32], LAPGAN [33], etc.

$$\min_{G} \max_{D} V(D,G) = E(x,z) \tag{3}$$

$$E(x,z) = E_{x \sim p_{data}}(x)[logD(x)] + E_{z \sim p_z}(z)[log(1 - D(z))] \tag{4}$$

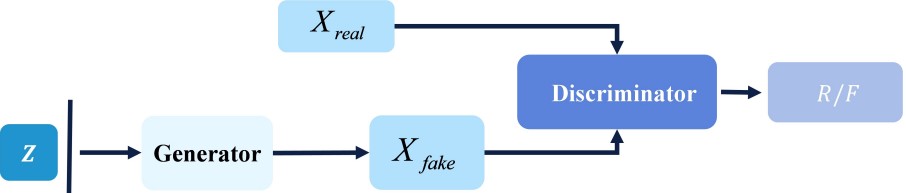

**Figure 8.** The model architecture of generative adversarial networks, where the generator generates fake data and the discriminator reasonably identifies the original data and the generated fake data.

In Computer Vision, GAN has demonstrated nearly omnipotent generation capabilities and a high degree of completeness for the job of text image generation. There is a method that uses the concept of confrontation learning to provide privacy-preserving text representation in the context of EHR [34]. Although the architecture of GAN is not directly used, the principle of the confrontation game contained in it was used. This survey will be briefly described in specific applications.

### 4.6. Graph Neural Network (GNN)

The Graph Neural Network (GNN) is a neural network that learns graph-structured data. The network has the feature of acquiring data with graph structure and can be applied to many fields such as clustering, classification, and generation. Early GNNs use RNN as the main structure, which simply generates vector representation for each graph node and does not have a good response to complex and variable actual graph structure data.

The Graph Convolutional Neural network (GCN) constructed by [35] combined with CNN for this problem. Furthermore, they also proposed spectral decomposition convolution and spatial graph convolution. The GCN, shown in Figure 9 and Equation (5), has two input components: the feature matrix $X \in R^{N \times F}$ and the adjacency matrix $A \in R^{N \times N}$, $N$ is the number of nodes in the graph structure and $F$ is the number of input features per node. $H_l$ is its hidden layer and $l$ represents the $l_{th}$ hidden layer. $W_l$ and $\sigma$ represent the weight matrix of layer $l$ and the nonlinear activation function of the network, respectively, and $W_l$ are shared among different graph nodes. Matrix $Z = H_N$ is the out-

put of this GCN. Because the network itself has properties that enable GCN to perform far better than other methods for tasks such as node classification and edge prediction, the techniques for processing and applying EHR data using graph neural networks have become widespread. Furthermore, there are some common graph neural networks, such as GRN [36], GAE [37], GAT [38], etc. Graph representation learning [39–42] corresponding to graph neural networks has been of great help for the play of deep learning in medical tasks.

$$H_{l+1} = \sigma(AH_lW_l) \tag{5}$$

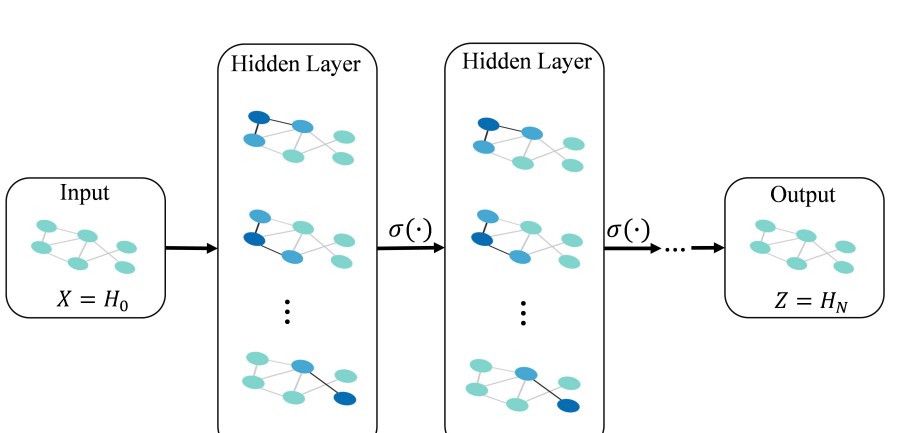

**Figure 9.** The model architecture of GCN.

## 5. Deep Learning Applications in EHR

In this section, we introduce how deep learning has been used on medical health records in recent years. Compared with the recent review [4] on the application of deep learning in EHR, this review also classifies the application of deep learning in EHR in terms of four tasks, i.e., *Information Extraction*, *Representation Learning*, *Medical Result Prediction*, and *Privacy Protection*. In addition, as shown in Table 2, these tasks are again classified in this review. *Information Extraction* includes named entity identification, relation extraction, and hidden data extraction. *Representation Learning* includes vector-based patient representation, patient representation based on time matrix, graphic-based patient representation, and sequence-based patient representation. *Medical prediction* includes static prediction and temporal prediction. *Privacy Protection* includes data de-identification with patient representation protection.

**Table 2.** Research Questionnaire on deep learning in the Field of EHR.

| Task | Model/Method | Publication Year | Proposed by |
|---|---|---|---|
| Named Entity Recognition | - | 2019 | Koleck et al. [43] |
| | - | 2019 | Sheikhalishahi et al. [44] |
| | - | 2019 | Savova et al. [45] |
| | Bio-BERTv1.1 (+PubMed) | 2020 | Lee et al. [46] |
| | Bidirection Language Modeling | 2018 | Sachan et al. [47] |
| | MTM-CW | 2019 | Wang et al. [48] |
| | MTL-MEN&MER feedback + Bi-LSTM-CNNs-CRF | 2019 | Zhao et al. [49] |
| | KA-NER | 2021 | Nie et al. [50] |
| | Bi-LSTM + CRF | 2020 | Wei et al. [51] |
| | DLADE | 2019 | Wunnava et al. [52] |

**Table 2.** *Cont.*

| Task | Model/Method | Publication Year | Proposed by |
|---|---|---|---|
| Relation Extraction | JOINT | 2020 | Wei et al. [51] |
| | SVM, Deep Learning model, Rule Induction | 2018 | Munkhdalai et al. [53] |
| | FT-BERT, FC-BERT | 2019 | Wei et al. [54] |
| | Hybrid CNN architecture with focus loss function | 2019 | Sun et al. [55] |
| Hidden Data Extraction | VRNN + VAE | 2015 | Chung et al. [29] |
| | VRNN + GRU-U | 2020 | Jun et al. [56] |
| Vector based patient representation | Fully Connected Neural Network | 2015 | Che et al. [57] |
| | LSTM,TANN, NN with boosted time based decision stumps | 2018 | Rajkomar et al. [58] |
| | CNN | 2016 | Cheng et al. [59] |
| | AE-based | 2016 | Miotto et al. [1] |
| | MCEMJ, MCEMC and MCECN | 2016 | Choi et al. [60] |
| Patient representation based on time matrix | CNN-based + NMF | 2016 | Cheng et al. [59] |
| | - | 2012 | Wang et al. [61] |
| | - | 2014 | Zhou et al. [62] |
| Graphic-based patient representation | Temporal Graph Model | 2015 | Liu et al. [63] |
| | GCT | 2017 | Choi et al. [64] |
| | KAME | 2018 | Ma et al. [65] |
| | CogDL | 2021 | Cen et al. [41] |
| | DIG | 2021 | Liu et al. [40] |
| Sequence-based patient representation | Med-BERT | 2021 | Rasmy et al. [66] |
| | GCT | 2020 | Choi et al. [67] |
| | Attention Model | 2018 | Song et al. [68] |
| | BEHRT | 2020 | Li et al. [69] |
| Static medical prediction | ANN + Linear model | 2016 | Choi et al. [70] |
| | enRBM | 2015 | Truyen et al. [71] |
| | AE-based + LR Classifier | 2016 | Miotto et al. [1] |
| | Deep EHR | 2018 | Liu et al. [72] |
| Temporal medical prediction | STAN | 2021 | Gao et al. [73] |
| | End-to-end DL + Classic ML | 2020 | Martin et al. [74] |
| | Multimodel based on LSTM and CNN | 2021 | Yang et al. [75] |
| | BiteNet | 2020 | Peng et al. [76] |
| Data de-identification | RNN | 2017 | Dernoncourt et al. [77] |
| | multiple RNN | 2016 | Yadav et al. [78] |
| | Stack RNN + attention mechanism | 2020 | Ahmed et al. [79] |
| Patient representation protection | Collaborative Privacy Representation Learning Method | 2020 | Festag et al. [80] |
| | Adversarial Learning | 2016 | Yadav et al. [78] |
| | Confrontational Learning Method & De-clustering | 2016 | Coavous et al. [81] |
| | Self-joining Adversarial Network | 2019 | Friedrich et al. [34] |
| | PATE-GAN | 2018 | Jordon et al. [82] |

*5.1. Information Extraction*

EHR data is often arranged in medical information systems for management, and clinical records are more thorough. Each patient's medical records will generate unique EHR data, which is frequently linked to a variety of clinical documents, including admission records, diagnostic records, doctors' summaries, and prescription lists. The unstructured section of EHR is extremely difficult to obtain relevant information. Prior to the adoption

of deep technology, these tasks were primarily performed manually, necessitating a significant amount of human feature engineering and ontology mapping, which is the primary cause for the restricted usage of relevant data information. In this part, we look at how two technologies of NLP methods (NER and Relation Extraction) can be used to extract EHR data.

5.1.1. Named Entity Recognition (NER)

The basic goal of the EHR entity recognition task is to recognize medical-named entities in medical texts, such as disease names, medicine names, and related technical semantic data.

Disease name: The basic purpose of NER in medical text information is disease name recognition. Medical publications and clinical reports make up the majority of the corpus. The disease-related text is mapped to the same semantic type by NER. Sheikhalishahi et al. [44] have explored the use of natural language technology in chronic disease clinical records, concluding that shallow classifiers and rule-based techniques are now the most popular. They also recognized that, while these algorithms can extract useful medical information from text, they essentially eliminated the ability to extract more intricate correlations from medical texts. Koleck et al. [43] focused on using natural language processing to assess the symptom information in the free text of EHR as a potent base for illness diagnosis, although deep learning technology is not frequently used in this field. Savova et al. [45] looked into how NLP is used in oncology and cancer characterization. Furthermore, computer scientists and medical professionals must collaborate to boost the usefulness and applicability of NLP technology in this industry from a disciplinary integration standpoint.

At present, the best performance in the recognition of the disease name recognition is close to 90% (F1-score). Lee et al. [46] proposed Bio-BERT, the first BERT model focused on specialized disciplines and demonstrated that pre-training BERT on the biomedical corpus considerably increased its performance. This model not only retains a high-level performance but it also has a considerable standard of generalization (only a little modification is needed for different tasks). Sachan et al. [47] have also conducted relevant research in this area, obtaining various degrees of neural networks based on general neural network improvement. The character-level convolutional neural network (CNN) layer, word embedding layer, word-level Bi-LSTM layer, decoder layer, and sentence-level label prediction layer are all part of the Bidirectional Language Model made up of these upgraded neural networks. Wang et al. [48] suggested a multi-task learning framework (Bi-LSTM + CRF) for Bio-NER to focus on the training data of various types of entities and improve each entity's performance. They argued that MTM-CW was the best-performing multi-task model since it shares the most information among tasks. Zhao et al. [49] also suggested a multi-task learning paradigm that can set up parallel multi-tasks while maintaining bidirectional task support. These studies have dealt with data scarcity to some extent; however, as indicated in Table 3, the best performance in specific task performance is still [46]. The recently proposed KA-NER by Nie et al. [50] incorporates a relevant knowledge base compared to most current NER models that focus only on contextual content or artificially added features. The KA-NER model performs well on multiple datasets and has a high generalization capability due to the average F1-score of 84.82% achieved by this model on multiple datasets.

As shown in Table 2, there are many indications in the evaluation of NER outcomes, and the scores obtained by the same model on different indicators may be inconsistent, and when utilizing different data sets, model performance may also improve or decrease. At the moment, the system architecture proposed by [49] has the most stable comprehensive score, and the systems demonstrated by [46] have the best performance of deep learning systems, with a performance gap of roughly 7.6% compared to the similar non-deep learning system.

**Table 3.** F1-scores of each model on different datasets.

| Method | Dataset | F1-Scores % |
|---|---|---|
| Bio-BERTv1.1 (+PubMed) [46] | NCBI Disease<br>2010i2b2/V A<br>BC5CDR | 89.7<br>86.73<br>87.15 |
| MTM-CW [48] | BC2GM (Exact)<br>BC2GM (Alternative)<br>BC4CHEMD<br>BC5CDR<br>NCBI-Disease<br>JNLPBA | $80.74 \pm 0.04$<br>$89.06 \pm 0.32$<br>$89.37 \pm 0.07$<br>$88.78 \pm 0.12$<br>$86.14 \pm 0.31$<br>$73.52 \pm 0.03$ |
| MTL-MEN&MER feedback + Bi-LSTM -CNNs-CRF [49] | NCBI Disease<br>BC5CDR | 88.23<br>89.17 |
| Bidirectional Language Modeling [47] | NCBI Disease<br>BC5CDR<br>BC2GM<br>JNLPBA | 87.34<br>89.28<br>81.69<br>75.03 |
| KA-NER [50] | NCBI-Disease<br>CoNLL03<br>Genia<br>SEC | 86.12<br>92.40<br>75.59<br>85.16 |

Medical related: Medical information, the second-largest task in the medical text NER, is more complicated than disease name because it includes several subtasks such as drug name, frequency of dispensing (Dr-Freq) and route of dispensing (Dr-Route), drug dose (Dr dose), duration of drug use (Dr-Dur), and adverse drug reaction events (Dr-ADE). These subtasks are closely tied to medical data and are based on specific standards and specifications, such as the international standard HL7 and the Fast Healthcare Interoperability Resources (FHIR). The NER task in clinical reports has a higher entity type complexity, but in medical reports or medical papers, NER must cope with distinct corpora and text types of disease recognition tasks. According to the survey, when both methods are used with the same corpus, the deep learning method performs better than the non-deep learning method.

In terms of the current performance of multiple NER models in medical health data, the i2b2/n2c2 corpus has practically all of the best results. Wei et al. [51] proposed a neural network architecture (Bi-LSTM + CRF) in the field of drug name recognition, which can model the target drug information and the associated sequence utilizing label characteristics, and the JOINT technique was derived on this premise. Using a new transformation, the JOINT approach generates drug-focused sequences from annotated texts. Some rules are changed in their study to boost recognition performance even further, and the F1-score is increased to almost 95% after postprocessing. There are several great models for determining drug distribution routes, and the difference between them and the estimated ideal system is only approximately 1%. The current performance in identifying hazardous drug response events is dismal, with a deviation of more than 10% from the predicted value, implying that identifying adverse drug reaction events is the most difficult task. On this data set, the F1-score of the DLADE system [52], which integrates Bi-LSTM and CRF into a deep network system, reaches 64 percent, whereas the F1-score of the model proposed on similar task [51] is 53 percent.

The F1-score generated by the deep learning model has reached an ideal range in additional tasks, such as drug frequency recognition, medication delivery route, and drug dose identification. In general, adverse drug response events are the most difficult types to adequately represent in plain language in NER tasks connected to drug information, which is why current deep learning in the subtask is unable to achieve good performance.

### 5.1.2. Relation Extraction

Relation extraction, in addition to entity recognition, is a significant task in the medical information sector. Its goal is to discover a hidden relationship between entities. The focus of this section is on the link between drugs and drug properties.

Earlier, Munkhdalai et al. [53] explored the performance of three machine learning methods for relationship extraction, namely SVM, RNN-based model, and Rule Induction. Their study confirmed that SVM was used for better performance compared to the other two models when the amount of data was small. However, when the amount of data in the training set grew, the performance of the deep learning network was much higher than that of the SVM model, with the F1 score of the LSTM increasing by almost 20% while the SVM only increased by 8%. Therefore, using deep learning for medical relationship extraction has great potential.

The research related to relationship extraction using deep learning is growing year by year with excellent results. The JOINT technique described by [51] based on CNN-RNN connected with postprocessing has great performance in the drug-relation extraction task, outperforming the basic CNN-RNN method. The former had an F1 score of 98.53 percent, while the latter had a score of 97.60 percent. The CNN-RNN architecture, which is more inclined to embed in the self-training domain, appears to be better than the Bi-LSTM architecture based on the attention mechanism in their research, as the F1-score of CNN-RNN architecture was 97.2 percent compared to the F1-score of Bi-LSTM architecture, which was 96.44 percent.

The best model for drug relationship extraction is the recursive hybrid CNN architecture with a focus loss function proposed by Sun et al. [55] in 2020, which has an F1-score of 78.3 percent, and the best model for drug attribute relationship extraction has an F1-score of 98.7 percent; the difference between these two models has exceeded 13 percent. Similar to the CNN-RNN model architecture, the greatest performance on this job is still CNN-RNN model architecture, with the exception that Bi-LSTM based on attention mechanism works exceptionally well.

Until recently, Wei et al. [54] introduced Bert into medical relationship extraction and constructed the FT-Bert model and FC-BERT based on the BERT model proposed by Devlin et al. [83] On the MIMIC-III dataset, the FT-BERT model achieved higher scores in different types of relationship extraction compared to the CNN-RNN and JOINT approaches mentioned above. Especially in the more difficult types of relation extraction where it scores higher by about one percentage point than the first two.

### 5.1.3. Hiding Data Acquisition

Hidden information acquisition is a natural language processing technology that has made significant advances in the field of medical data. E. Jun et al. [56] proposed the VRNN network type, which is coupled by GRU-U and can infer and compensate for missing information in medical text data. Chung et al. [29] proposed VRNN, which uses the model architecture of VAE to give the network the ability to infer information. In the prediction challenge, the model architecture's AUC score was 83.2 percent. While this technology can assist users in utilizing medical texts that are lacking in information, it also introduces new issues: attackers can gain illicit information by mining potentially hidden information in data, which may be a concern that needs to be addressed in the near future.

### 5.2. Representation Learning

The essence of feature learning in medical health records is representation learning for patients, which must be learned from EHR to encode relevant information. Due to the high dimensional sparsity of EHR data, this endeavor typically employs the deep learning method for data encoding. This section will go through five commonly utilized methods.

### 5.2.1. Vector-Based Patient Representation

The goal of vector-based patient feature representation is to turn each patient's data into a mathematical vector that can be made up of completely linked neural networks, CNN, AE, or Word2Vec.

At an early point, Che et al. [57] attempted patient representation learning using fully connected neural networks, and most contemporary investigations used the framework proposed by Rajkomar et al. [58] as a baseline model. This represented technique sorts the complete EHR data by time, and the data is sorted both by the patient and by time. To label the data in each attribute, this model employs three neural network methods (LSTM, TANN, and a neural network with boosted time-based decision stumps). This data structure is much more straightforward. Although the produced data can be utilized for any prediction challenge, the created features may or may not perform better on certain tasks. The CNN network developed the Text-CNN network, which is specifically suited to handle text data. It obtains patient characterization in several convolution-processing forms, with more detailed patient attributes. In addition, attempts have been made to include temporal information outside the convolution layer to imitate patient longitudinal information in CNN's more complex variation networks [59].

Above, the Auto-Encoder(AE) was briefly described. Denoising AE [84], Stacked Denoising AE [85], Variational AE [86], and Compressed AE [87] are all variations of the vanilla Auto-Encoder. Miotto et al. [1] used AE for patient representation learning for the first time, based on its great performance in representation learning. An unsupervised three-layer stacked automated denoising encoder is the model architecture proposed in this paper. This model can be used to create a detailed and versatile patient portrayal.

Word2Vec is a popular approach for large-scale natural language word embeddings. In Word2Vec, there are primarily two algorithms: CBOW and skip-gram. Both methods have their merits, but only one hidden layer is present in each. As a result, they are known as shallow neural networks. Variants have recently been applied to patient representation learning problems, such as the technique proposed by Choi et al. [60] that used three types of embeddings: MCEMJ, MCEMC, and MCECN. The link between coding sequences is also among the features obtained.

### 5.2.2. Patient Representation Based on Time Matrix

The two-dimensional matrix created by the patient representation based on the time matrix has one dimension of time and the other dimension of EHR clinical events. Non-negative Matrix Factorization (NMF) is the fundamental algorithm involved, which may break down data made up of a set of non-negative elements into high-dimensional data. Early studies [61,62] speculated on the feasibility of encoding the latent factors of ad hoc patient data through a one-to-one identifiable mapping between patient representation matrix and target labels. The CNN-based model proposed by Cheng et al. [59] attempts to use a variant of NMF for patient representation learning in EHR and analyzed the special properties of EHR data itself, such as sparsity and temporality.

### 5.2.3. Graphic-Based Patient Representation

Each patient can be represented as a compact graph using the graph-based patient representation. The edges between nodes in the graph encode the association between clinical events, while the nodes themselves encode clinical events. For instance, Liu et al. [63], the first to learn graph representation, provided a temporal graph model that may give sufficient information for various analysis tasks. Graph representation has made significant strides in deep learning and evolved into a Graph Neural Network (GNN), despite the fact that the technique in this study is not based on deep learning technology. The use of GNN in EHR improves the model's interpretability and efficiency. Choi et al. [64] introduced the Graph Convolutional Transformer (GCT), which is based on a transformer and data statistics and can narrow the search space to only those areas that exhibit significant attention distributions rather than the full space. Ma et al. [65] also proposed KAME,

a knowledge-based attention model that can fully utilize information in medical texts, increase downstream task performance, and obtain both accurate embedding of medical code and knowledge in prior code. They both use a hierarchical ICD-9 knowledge graph as a graph model, which is the same.

Recently, many efficient open-source graph representation learning platforms have emerged. Cen et al. [41] proposed the CogDL system, which includes graph representation learning and GNN. researchers can quickly build their own experiments through this system. The DIG system proposed by Liu et al. [40] also has interpretable analysis and deep learning of 3D graphs compared with CogDL, which further advances graph representation learning.

### 5.2.4. Sequence-Based Patient Representation

The event sequence feature with a time stamp is the sequence feature generated by patient representation based on the sequence. For sequence-based patient representation learning, RNN and its derivatives (GRU, LSTM, etc.) are frequently utilized. Although RNN and its variants can do sequence modeling, especially LSTM, such networks have drawbacks when dealing with extended sequences. Although a two-way network design (such as Bi-LSTM) already exists, it increases training time and the two-way structure is just a two-way link, rather than parallel training.

As a result, new research has begun to develop the transformer architecture, which includes a self-attention mechanism and embedding of location information. This paradigm is capable of valid bidirectional representation. Rasmy et al. [66] have used it to learn patient representations and model clinical sequences. They exhibited a Med-BERT design that uses the same transformer framework as the original BERT. Moreover, numerous other researchers have studied it [67–69]. The transformer's training process is comparable to that of employing a neural network to model EHR data sequences, and it also encodes each clinical event matching to each timestamp as a unit, as well as the entire patient information as a full sequence. This differs from the previous variant in that RNN predicts the next output via recursion. The transformer accepts the unit sequence as a whole and learns basic information from it using a self-attention process.

### 5.3. Medical Prediction

Medical prediction is currently the main application of deep learning in EHR, and the performance of neural networks trained on big data is usually satisfactory. Although there are many different prediction objects, medical outcome prediction can be broadly thought of as static prediction and temporal prediction.

### 5.3.1. Static Medical Prediction

A variety of prediction tasks are included in static medical prediction. One of the simplest categories is the prediction of some outcome without considering time constraints. For example, in heart failure prediction studies, early methods mainly used KNN [88], Logistic Regression (LR) [89,90], SVM [91] and MLP [92]. Choi et al. [70] introduced the conversion of data into vectors based on conventional methods and used distributed representation methods with several ANN and linear models for prediction.

In addition, Truyen et al. [71] proposed an enRBM model to perform prediction tasks. This model is capable of fast characterization learning for large data with high dimensionality compared to previous models and can support various medical analysis tasks. For example, they plugged in a logistic regression classifier after the model to perform suicide risk stratification, and in further experiments demonstrated that the results obtained using the full EHR data were superior to those using only diagnostic codes. In addition, Mitto et al. [1] proposed Deep Patient using AE-generated data vectors to access an LR classifier for various ICD-based disease diagnoses and obtained high k-values for the precision@k metric.

In addition to the data prediction performed using the above, deep learning has been able to use medical texts in EHR for various types of prediction due to the rich medical information contained in EHR. Medical texts are classified into structured texts and unstructured texts in the same way as conventional texts. Liu et al. [72] proposed a general multi-task framework for disease onset prediction based on previous studies [93–95]. This framework does not require special feature engineering and can utilize both structured and unstructured data compared to traditional methods.

### 5.3.2. Temporal Medical Prediction

In contrast to static medical prediction, this type of prediction task focuses on processing data based on time series and predicting the outcome or predicting the occurrence of a disease.

Martin et al. [74] proposed a framework of end-to-end DL + classic ML to perform Chronic Heart Failure (CHF) detection using heart sound data for the first time. The accuracy of the method reached 92.9%, which is already very close to the accuracy of the manual assessment. The STAN model using GCN for Spatio-temporal modeling proposed by Gao et al. [73] was tested on COVID-19 data using real data for epidemiological prediction. According to the task, they performed graph construction, using location information as graph nodes to construct graph edges with geographic proximity and node population size, and fusing static features with dynamic node features in the graph nodes. The results are better compared with the epidemiological model SIR, SETP, and other deep learning algorithms [28,96,97].

Moreover, mixing temporal information with textual data is becoming a popular choice to improve the performance of medical prediction tasks. Yang et al. [75] proposed a multimodal architecture for mortality prediction. This architecture LSTM network performs representation extraction of time series, text embedding of text data from clinical records with CNN, and finally, fuses the feature vectors of both types for prediction. This multimodal model performed much better on the mortality prediction task than the model using a single data type, scoring eight percentage points higher on AUCPR than the LSTM and 0.3 percentage points higher than the previous multimodal model Multi-CN. Peng et al. [76] proposed BiteNet with a self-attentive module called Masked Encoder (MasEnc). This special self-attentive mechanism would extract contextual and temporal relationships from the medical text. Compared with previous work [68,98–100], BiteNet achieves better prediction performance on long sequences of medical texts.

### 5.4. Privacy Preservation

As the principal technology for protecting patients' personal information, data de-identification technology has played a part. However, the medical data that is being de-identified is not completely secure. Furthermore, some researchers will use the EHR data set they produced for connected research due to the applicable provisions. Such information is likely to be used in research without being de-identified, exposing patients' private details. The concealed information can currently be found mostly in the trained patient depiction and the data itself. Adding noise or cover to the data is a common technique, but it lowers the accuracy and performance of downstream activities. As a result, we looked at the latest revolutionary technologies for protecting patient privacy.

### 5.4.1. Data De-Identification

The patient's personal health information is frequently included in medical data. HIPAA requires that all released clinical data be stripped of sensitive information such as the patient's name, social security number, and information about the hospital, region, and time, making public EHR data difficult to publish. There are additional bills in the works in the United States to secure patients' personal information and prevent it from being misused. Despite the fact that rigorous legislation can ensure the confidentiality of patient information, data processing will almost certainly require a lot of manual interaction.

The early automatic data de-identification techniques [101–103] used were implemented by designing corresponding algorithms for PHI or other specific private data, which lacked generalization capability and required a lot of human work in the early stage. Neamatullah et al. [104] proposed a de-identification algorithm with some generalization capability. However, the algorithm is still based on heuristic pattern matching with dictionary lookup, which is general enough but far from enough to support the development of open medical data.

As a result, Dernoncourt et al. [77] presented an automatic system for clinical text identification, which would eliminate the need for time-consuming manual intervention. Bi-LSTM, as well as character and word embedding, make up the framework. This method is currently quite advanced, according to the investigation. Furthermore, for automatic identification, the integrated method of conditional random fields can be used. Meanwhile, in the field of recognizing possibly recognizable named entities in clinical literature, Yadav et al. [78] investigated the combination of multiple RNN variants with various word embedding models and found that all RNN variations outperform the typical baseline approach.

Ahmed et al. [79] presented the first recurrent neural network with an attention mechanism based on [77]. The network further enhances the de-identification ability by stacking recurrent neural networks and adding the attention mechanism, which is about 0.03% higher than [77] in the F1-score. However, the number of parameters of the model increases dramatically and is three times higher than [77], reaching 110,000,000.

### 5.4.2. Patient Representation Protection

In most cases, trained patient representation is required before the following procedure (such as disease prediction). This representation contains a lot of information about patients, and attackers can use it to obtain confidential information. Yadav et al. [78], which is inspired by Goodfellow's adversarial learning, adds a discriminator after representation learning, intending to obtain a patient representation that can predict tasks effectively but cannot detect private information. The final objective function is shown in Equation (8), where $X$ is the cross-entropy function, $x$ is the input, $y$ is the prediction goal, and $b_i$ is the prediction objective of the $i_{th}$ privacy information. The joint learning is formed by the model parameter theta $M$ and the identifier parameter $\theta_D$.

$$\hat{\theta} = \min_{\theta_M} \max_{\theta_D} = X(\hat{y}(x; \theta_M), y) - \sum_{i=1}^{N} \lambda_i X\left(\hat{b}(x; \theta_{D_i}), b_i\right) \tag{6}$$

The model parameter $\theta_M$ and the discriminator parameter $\theta_D$ are utilized to construct the joint learning utilizing the prediction target of privacy information. Friedrich et al. [34] also exploited the adversarial concept, dividing the network into two sections for alternate training utilizing self-joining adversarial network simulation attacks. To begin, the de-recognition and representation models are pre-trained simultaneously to optimize the de-recognition loss value $l_{dided}$. The second step is to freeze representation model training for confrontation model training and maximize the confrontation model's loss value $l_{adv}$. The alternate training of the two continuously improves the overall system's ability to resist attacks, and the overall system's F1 scores are above 96 percent. Coavoux et al. [81] studied two protective measures: the confrontational learning method and de-clustering, and verified the multi-corpus model. On any corpus, the model outperformed the baseline model by a wide margin. Jordon et al. [82] proposed PATE-GAN with differential privacy based on DP-GAN [105]. The model protects the information of the original data while ensuring the availability of the data. The model outperforms DP-GAN with the same privacy parameters by an average of 5%. PATE-GAN is also able to limit the impact of individual data on the model to ensure the performance of the model.

In addition, the researchers consider the fact that in a real-world environment, many healthcare organizations intelligently train locally using a small amount of data, and data sharing is needed to have enough training data. Thus, a distributed training approach

using patient representations extracted by neural networks is proposed. However, in this process, there is still a possibility that the representation information can be recovered to the original data [106]. Festag et al. [80] combined the parallel training method proposed by Shokri et al. [107] with the de-identification network proposed by Dernoncourt et al. [77] to design a collaborative privacy representation learning method to minimize the risk of PHI leakage.

## 6. Challenges for EHR

Information extraction, representation learning, result prediction, data de-identification, and privacy information protection were among the applications in the EHR application based on deep learning that were reviewed. Although the application of deep learning in EHR has made great progress, especially in medical prediction tasks, the performance of deep learning models has reached a satisfactory level. However, the shortcomings of deep learning in this field still exist. This review analyzes the technologies that still have the potential for development in the above research, such as information extraction, representation learning, and private information security, using inquiry and statistics.

### 6.1. Limitations in Representation Learning

Currently, representation learning is constrained by the lack of datasets and the weak generalization ability of the models. When the corpus is fixed, representation learning has a significant impact on the model's end performance at the starting point of applications. As previously stated, there are various models to choose in representation learning, and appropriate approaches can be chosen depending on the data type. Corpus selection has a significant impact on feature learning. EHR data that can be reviewed by the public is still rare at the moment. Many researchers rely on self-collected EHR data sets since they are unable to meet the requirements of the public release of EHR data, implying that open EHR data still need growing. Representation, on the other hand, usually has two opposing characteristics: universality and distinctiveness. The outcomes are poor if representation is universal, despite the fact that it can be employed reasonably in many downstream tasks. Most researchers in various deep EHR research topics have particular task-specific training to assure models' performance. Alternatively, you may use transfer learning to retrain based on prior information in order to obtain the best feature vector for the model. The fundamental source of these algorithms is natural language processing. Hence, the current rapid growth of NLP technologies is bound to hasten the development of deep learning on EHR.

### 6.2. Deficiencies in Information Extraction

Many subtasks in information extraction have achieved good performance, but flaws remain. For example, when it comes to the discrimination of adverse medication reaction events, survey respondents might deduce that in order to accomplish the work successfully, they must address the complex interaction between medical events, data access, and the time–space relationship. In general, extracting the intricate relationships in the medical literature is a promising area of research. In NLP, there have been numerous studies on extracting relationships between texts, particularly in long texts. Researchers can use current NLP research results to graft it into medical texts after taking into account the unique characteristics of medical text content.

### 6.3. Prospect of Patient Privacy Protection

De-identification technology and rigorous legislation can ensure that the personal health information of the patients involved is not exposed prior to the data being published (while also restricting the amount of public data sets). This information is only available in text format. The usage of neural network technology can still infer privacy information from identity data at the moment. After further examination, it is clear that Goodfellow's concept of building adversarial networks is useful. Many studies have been prompted to

investigate how to improve the security of models by simulating attackers. Deep learning is currently in full bloom, and a network with inference capability is progressively evolving. As previously stated, the VRNN-VAE model was able to extract missing data from the network. Although this raises the likelihood of optimal utilization, it also introduces new difficulties to patient information security. That is, an attacker can extract concealed information from a text by leveraging the neural network's reasoning capacity, which is akin to human thinking and decryption of the text to reach conclusions. The current protection method is mostly based on the confrontation mechanism, with the protection object being the feature representation acquired by feature extraction. However, this mechanism's protection capabilities are still being refined. Second, the mysterious nature of neural networks has a significant impact on the practical implementation of privacy protection technology.

### 7. Conclusions

At present, various models and algorithms of deep learning have been widely applied to EHR, which can effectively accelerate the solution of various medical tasks. Deep learning in EHR is a new field compared to other deep learning applications such as Computer Vision (CV) or Natural Language Processing (NLP). This paper composes and outlines the relevant research based on EHR and deep learning in recent years, and describes the development of deep learning in EHR from four application areas, namely information extraction, representation learning, outcome prediction, and privacy protection, respectively. In addition, this survey also provides an overview of common deep learning models in various application areas of EHR. The common English EHR datasets that have been publicly available are also summarized. By analyzing the recent research status in this area, this paper identifies the problems and challenges that still need to be addressed in the existing areas of representational learning, information extraction, and privacy preservation.

Currently, there are still many challenges for EHR applications based on deep learning. As described in this paper, there are many similarities between the field and NLP, which means that it is possible to migrate techniques from NLP to EHR. For example, in representation learning the representation extraction model in NLP can be redesigned according to the characteristics of medical data itself. When performing medical privacy protection, in addition to using the adversarial algorithms used in the existing work mentioned in this paper, the protection algorithms in CV can also be improved for textual data protection. With the gradual progress of deep learning in EHR research, deep learning-based EHR will be widely promoted in terms of performance and security, which will bring great convenience to healthcare services.

**Author Contributions:** Conceptualization, J.X. and X.X.; methodology, J.X.; validation, X.X., J.C. and V.S.S.; investigation, J.X.; resources, X.X.; data curation, J.X.; writing-original draft preparation, J.X. and X.X.; writing-review and editing, J.X., X.X., V.S.S., J.M. and J.C.; visualization, J.X.; supervision, X.X. and J.M.; funding acquisition, X.X. and Z.C. All authors have read and agreed to the published version of the manuscript.

**Funding:** This research has been supported by the National Natural Science Foundation of China under grants 61876217 and 62176175; the Innovative Team of Jiangsu Province under grant XYDXX-086; the Science and Technology Development Project of Suzhou under grants SGC2021078.

**Institutional Review Board Statement:** Not applicable.

**Informed Consent Statement:** Not applicable.

**Data Availability Statement:** Publicly available datasets were analyzed in this survey. These data can be found here: MIMIC, https://mimic.mit.edu/; eICU-CRD, https://eicu-crd.mit.edu/; PCORnet, https://pcornet.org/; Open NHS, https://digital.nhs.uk/; NCBI-Disease, https://www.ncbi.nlm.nih.gov/CBBresearch/Dogan/DISEASE/; i2b2/n2c2 NLP Research Data sets, https://www.i2b2.org/NLP/DataSets/. All these links are accessed on 16 November 2022.

**Conflicts of Interest:** The authors declare no conflict of interest.

**Abbreviations**

The following abbreviations are used in this manuscript:

| | |
|---|---|
| KNN | K-Nearest Neighbor |
| LR | Logistic Regression |
| SVM | Support Vector Machine |
| EHR | Electronic Health Record |
| NLP | Natural Language Processing |
| CV | Computer Vision |
| ANN | Artificial Neural Network |
| ML | Machine Learning |
| SIR | Susceptible-Infected-Removed |
| SEIR | Susceptible-Exposed-Infected-Removed |

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
