# Peer review of "A Survey of Deep Learning for Electronic Health Records"

_applsci, doi:10.3390/app122211709_

Round 1
Reviewer 1 Report
I think when talking about medical data one has to put emphasizes on the fact that medical data is strongly coded using specific formats e.g. HL7, coding systems e.g. ICD, ATC, LIONC, SNOMED etc as well as related value sets.
HER was used instead of EHR
Author Response
Thank you very much for your valuable comments. We have carefully revised and added details to our manuscript. The PDF we have uploaded contains our detailed response. Please see the attachment.

Reviewer 2 Report
This is an interesting study and the authors have reviewed several EHR-related deep-learning applications. The paper is generally well structured. However, the paper has some shortcomings in regard to not only reviewing but also realizing some original data analyses, and also the contribution of the paper is not discussed in the conclusion and remains vague.
Author Response

(The authors gave the same response as above.)

Reviewer 3 Report
In my view, any survey paper should have at least 100 references and a detailed study of the chosen topic should be discussed for understanding the recent developments in the area will be helpful for all the researchers working in this area.
This work should discuss how the references are chosen and also elaborate on this presentation with more details on recent works.
This work needs significant improvement with more details to be added. A survey paper can include the following aspects and the authors can consider including some of the suggestions given below:
Evolution of the area
Comparison with existing surveys
Comparison between the current and the existing surveys in literature.
List of algorithms present for solving the literature problem:
Discussion on various applications pertaining to the broad application
considered
Recent advancements in the field
Roadmap and open issues
Detailed Future research directions
Outcome of survey
What are the novel steps taken by the authors to evaluate the popular existing models?
Suggestions for the future researchers.
Authors have to refer to some of the survey papers to incorporate significant changes into their work to have a full-fedged survey paper.
Author Response

(The authors gave the same response as above.)

Reviewer 4 Report
1- The authors need to update the refence list with the most recent works.
2-The authors should summarize the limitations and the contributions.
3-The authors should add others model of deep learning.
4-The authors strongly encourage to compare the contributions for each works in this paper.
Author Response

(The authors gave the same response as above.)

Round 2
Reviewer 2 Report
I consider that the revised version has addressed properly all comments for improvement.
Reviewer 3 Report
The comments are well addressed and this work can be accepted now.